# Religious Crisis as an Independent Causal Predictor of Psychological Distress: Understanding the Unique Role of the Numinous for Intrapsychic Functioning

**Jesse Fox** [1,*] and **Ralph L. Piedmont** [2]

1   Department of Counselor Education, Stetson University, DeLand, FL 32723, USA
2   Center for Professional Studies, Timonium, MD 21093-9998, USA; ralphpiedmont01@gmail.com
*   Correspondence: jfox2@stetson.edu; Tel.: +1-386-822-7132

**Abstract:** Religious and spiritual (R/S) struggles are tensions or conflicts one experiences in relationship to what is considered sacred or transcendent. In this study, we tested competing causal models of psychological distress as it relates to personality and R/S struggle using structural equation modeling. The study sample consisted of 226 (72.0%) females and 88 (28.0%) males ($n$ = 314) drawn from the Amazon's Mechanical Turk (MTurk) worker population. We found that though the five-factor model (FFM) of personality was a robust predictor of psychological distress, the R/S struggle added significant, incremental predictiveness. SEM analyses supported our contention that R/S struggle may represent a new, causal pathway of psychological distress that is *independent* from the FFM. Our findings are taken as evidence that R/S struggles require unique ways of conceptualizing their causal impact on clinical impairment and that psychological interventions need to systematically address numinous constructs in order to ensure that all aspects of emotional dysphoria are considered and their influences treated.

**Keywords:** R/S struggle; psychological distress; five-factor model of personality

## 1. Introduction

Religious and spiritual (R/S) struggles are tensions and conflicts between oneself and what one considers sacred or transcendent (Exline 2013; Pargament et al. 2011; Piedmont 2020). A growing body of evidence supports the link between R/S struggles and psychological distress (Exline 2013). There is also growing evidence revealing a connection between R/S struggles and personality (Wilt et al. 2016). However, the important question is how best to conceptualize the antecedent and consequent relationships between R/S struggles, personality, and psychological distress? Given the powerful predictive value of certain personality models, especially the five-factor model (FFM), on a variety of health outcomes, including psychological distress, it is important to establish whether or not R/S struggles uniquely contribute to our capacity to predict psychological distress beyond that which is better explained by personality (Wilt et al. 2016). Otherwise, criticisms that psychospiritual constructs are nothing more than the "religification" of psychological dimensions will remain unaddressed (Piedmont 2014; Van Wicklin 1990). More specifically, if R/S struggles are better explained either by the FFM or even as a consequence of psychological distress itself, then it is reasonable to argue that researchers and clinicians would be misplacing their focus by addressing R/S struggle directly, when they are better conceptualized as by-products of these underlying psychological pathways. Recent scholarship has begun to address these methodological concerns (see Joshua et al. 2018; Piedmont and Wilkins 2020). The current investigation extends previous research on R/S struggle by testing the viability of a conceptual model that understands R/S struggles as having a unique, causal impact on psychological distress that is independent of personality.

*1.1. A Brief Overview of the Origins and Psychological Significance of R/S Constructs*

While there is great interest in R/S constructs and their potential role in the mental life of individuals, there has been a chronic lack of any overarching psychological models which provide conceptual insights into the nature and function of R/S constructs (e.g., Gorsuch 1984, 1990; Piedmont 2014; Piedmont and Wilkins 2020). The scientific study of any phenomena has four goals in mind: (a) describe the construct, (b) predict the construct, (c), determine its antecedent and consequent relationships to other constructs, with the final objective in mind of (d) explaining the construct (Heppner et al. 2016). Whereas individual variables may be of interest, without either any conceptual (i.e., ontological) models that explain how and where these constructs emerged, and describe their nature (e.g., as motivations, cognitions, learned sentiments), or the identification of appropriate methodological models needed to develop and test them and determine their causal role(s), there is ultimately little the research literature that can provide professionals in their efforts to understand and apply these constructs coherently (Funder 2002; Hill et al. 2000; Koenig et al. 2012; Piedmont 2014). Koenig (2008) is more strident in his criticisms of the field when he noted, "Either spirituality should be defined and measured in traditional terms as a unique, uncontaminated construct, or it should be eliminated from use in academic research" (p. 349).

The lack of overarching psychological models to account for and predict the unique contribution of R/S constructs poses significant threats to any form of psychotherapy that may be called spiritually or religiously integrated (Captari et al. 2018; Pargament et al. 2011; Stewart-Sicking et al. 2019). Without a coherent framework, it is very difficult to explain to one's patient(s), one's stakeholder(s), or to one's ethical conscience why a particular approach is worth research or clinical time and effort. Though the rise of multicultural theory into mainstream psychological care has created a context for bridging the salience of R/S under this greater canopy of cultural relativity, current multicultural models lack the conceptual scope to adequately address the unique origins and consequence of R/S constructs. Stewart-Sicking et al. (2019) put the dilemma this way:

> In the end, the pluralism that R/S (religion and/or spirituality) force us to encounter is to accept paradox in a way that most multicultural competencies do not ask. I can accept rather easily that eye contact is seen as disrespectful in another culture; if I were a committed atheist and want to affirm that my client experiences the truth and not delusion through Islam, I would have to come to terms with a paradox. (p. 130)

Multicultural theory provides an initial professional validation for R/S constructs but does not go far enough in providing a psychological framework for clinically managing them.

Targeting this problematic lack of a coherent conceptual framework, Piedmont (Piedmont 2015; Piedmont and Wilkins 2020) has developed an ontological model that examines the physical origins of R/S constructs as well as discusses their psychological nature and importance for functioning. This model is based on three observations: (a) R/S are universal dimensions of human functioning, being significant factors in all cultures and across all ages; (b) there are individual differences in the extent to which individuals are sensitive to these factors, indicating that these dimensions are both inherent to people and evidence individual-difference properties akin to other personality-type constructs; and (c) R/S dynamics are unique to the human species; there are no animal models for these constructs. The existence of these qualities is rooted in the core of our humanity. Piedmont argued that R/S constructs (or what he refers to as numinous motivations) find their origins within the neo-cortex, that aspect of brain functioning that is responsible for the unique cognitive powers that characterize humans. For Piedmont, the numinous is a strictly psychological construct. It may include other phenomena such as mysticism and transcendence, but it represents only psychological qualities that are hypothesized to uniquely define the human species (Piedmont forthcoming; Piedmont and Wilkins 2020). While it may include the notion of a transcendent being, numinous constructs have no theological pedigree. These motivators are what make R/S so important to all humans. It also provides an understanding of R/S constructs that promotes interpretive value and clinical significance.

Piedmont identified three core numinous motivations: infinitude (I; our need to find personal durability for our strivings in life); meaning (M; the need to develop purpose and direction for our lives); and worthiness (W; finding personal acceptance of self within a transcendent perspective). What is important is that all of these qualities are strictly psychological in nature. There is no need for appeal to specific theologies or religious denominations in order to understand the value and role of these constructs. As noted above, our numinous motivations are what make R/S so important to us as a species. Having this ontological model provides direction and clarity to defining what is and is not a R/S construct (see Piedmont and Wilkins 2020 for a complete presentation of this model and its clinical applications).

*1.2. Conceptualizing R/S Struggles as Numinous Motivations*

While the neo-cortex in human beings provides unrivaled intellectual capacities to imagine, create, construct, and manipulate the world in ways that lend incredible adaptive capacities to our lives, it also creates new existential issues that our species must address. Specifically, human beings are the only species that knows from the beginning that they will one day die: life is temporal and limited. This finitude is an issue all people must deal with throughout our lives. Awareness of finitude creates psychological distress (Solomon et al. 1991). In response, human beings find ways to bring meaning, coherence, and depth to this transient life. The numinous constructs are those that enable us to live our lives productively, richly, and with a sense of personal value and satisfaction despite our awareness of finitude (Piedmont and Wilkins 2020). Thus, any model of psychological functioning needs to include these numinous motivations if it wants to be comprehensive and ecologically valid.

In Piedmont's (2020) original model of the numinous, The Assessment of Spirituality and Religious Sentiment (ASPIRES), religious crisis was a brief measure of R/S struggle, defined as the tensions or conflicts in one's relationship to the transcendent, primarily characterized by a sense that one's God is angry with or disapproving of them, as well as a sense of alienation from and a lack of acceptance in one's faith community. Elevated scores in religious crisis predict greater psychological distress (Piedmont 2020). In their refinement of this model, Piedmont and Wilkins (2020) recast religious crisis as one form of low W. They have shown how low scores on W were uniquely related to depressive symptoms, negative affect, and lower levels of resilience, self-compassion, and work satisfaction. Numinous motivations, including W, provide the core to human aspirations and their perceived place in the world. Importantly, these forces are independent of all other motivational variables, although they do constructively engage with all other psychological aspects of the person. Piedmont has shown that the numinous constructs are independent of existing personality models (e.g., the five-factor model (FFM) of personality), evidence incremental validity in predicting a wide array of salient outcomes over the FFM (e.g., well-being, meaning making, life satisfaction, coping ability, etc.); is recoverable across different data sources (e.g., self-reporting and observer ratings), generalizes across various languages, cultures, and religious denominations, and exerts an independent causal impact on psychological functioning (Piedmont and Wilkins 2020 provides reviews of all this research). Of particular interest, Piedmont et al. (2007) demonstrated that low scores on a measure of the W dimension were associated with characterological impairment, even after scores on the FFM were controlled. What can be concluded from these findings is that experiencing a lack of W creates existential struggles that are independent of the FFM.

This model understands numinous motivations as exceptional psychological variables, representing dimensions that uniquely define the human experience. As such, these constructs have little in common with basic personality dimensions (like the FFM domains, which are evident in non-human species, Gosling et al. 2003) and represent independent psychological forces that are inherent to higher order functioning. Consequently, this model postulates that impairments within our numinous motivations can have serious consequences for psychosocial functioning. These dysphoric influences operate independently from other aspects of personality (e.g., neuroticism).

Therefore, impairments in one's numinous motivations create a new pathway for the development of psychopathology.

### 1.3. R/S Struggle and Psychological Distress

R/S struggles may take on a variety of forms. Researchers have developed models of R/S struggle operationalized in terms of multiple dimensions (Exline et al. 2014), negative religious coping (Pargament et al. 2011), spiritual dryness (Büssing et al. 2013), and religious crisis (Piedmont 2020). A consistent line of research has documented a positive relationship between R/S struggle and psychological maladjustment. Research has found that R/S struggles are related to depression, anxiety, and global distress (Ano and Vasconcelles 2005), difficulty in adjusting to war-related trauma (Witvliet et al. 2004), suicidal ideation and behavior (Exline and Yali 2000; Piedmont forthcoming), among a host of other problems (Exline 2013). These findings remain consistent when examining clinical and non-clinical samples. For example, in a random sample from the general U.S. population, measures of R/S struggle positively predicted paranoid ideation, obsessive compulsions, depression, somatization, and anxiety (McConnell et al. 2006). Likewise, in a sample of psychiatric patients diagnosed with bipolar or schizophrenia, R/S struggles were positively associated with global psychological symptoms at baseline and one year later (Phillips and Stein 2007). In a sample of Roman Catholic Priests, R/S struggles were strongly correlated with depression, anxiety, and stress as well as burnout (Büssing et al. 2013). There are also linkages between R/S struggle and poor physical health outcomes. For instance, R/S struggle was associated with poorer health outcomes, cross-sectionally as well as longitudinally, in medical rehabilitation samples (Fitchett et al. 1999), stem-cell-transplant patients (Sherman et al. 2009), and trauma populations (Pargament et al. 1998).

This body of evidence has persuaded some scholars to conclude that R/S struggles predict lower psychological and physical health outcomes (Wilt et al. 2016, p. 342). However, to what extent R/S struggle continues to be associated with and to predict psychological distress once personality is controlled is less established. How R/S struggle may predict psychological distress using analytic procedures that go beyond correlation is also less known. Some exceptions to this general trend in the literature do exist. Wilt et al. (2016) recently studied the role of sacred moments in the context of R/S struggles, and using growth-curve modeling found that encountering sacred moments facilitated spiritual growth even in the context of experiencing R/S struggles. Harris et al. (2012), using longitudinal data, found that R/S struggles were moderately positively associated with post-traumatic stress disorder (PTSD) symptoms and mediated the relationship between baseline and follow-up symptoms over a one-year period of time. The magnitude of the mediation indicated that for every unit increase in Time 1 PTSD symptoms there was 0.10 increase in Time 2 PTSD symptoms that was mediated by R/S struggles. In a nationally representative sample, Pomerleau et al. (2019) found that R/S struggles significantly mediated the relationship between stressful life events and psychological adjustment, and that 47% of the effect was mediated by R/S struggle. Consistent with previous findings, R/S struggles predicted higher levels of negative psychosocial adjustment. While these studies with more methodological rigor support the theory that R/S struggles represent unique pathways of causing psychological distress, the replication and extension of these findings is still needed. Of importance to such research is identifying the direction of causality: do R/S struggles cause psychological dysphoria or vice versa? This is the key question that this report addresses.

### 1.4. R/S Struggle and Personality

As scholars have argued, it is important to understand how personality relates to R/S struggle because the way people generally interact with their environment will inevitably involve ways they respond to R/S struggles (Piedmont and Wilkins 2020; Wilt et al. 2016). Furthermore, though a significant amount of attention has focused on how personality relates to R/S, less scholarship has delved into how R/S struggles influence psychological functioning independently of personality. Some of the initial findings, however, are noteworthy. Most often, the FFM of personality is used as both

a predictor and a control variable of R/S struggles and scholars have found that several dimensions of the FFM are associated with the R/S struggle (Grubbs et al. 2016; Wilt et al. 2016). Neuroticism (N), a propensity toward negative affective states and emotional instability (Widiger and Costa 2002), appears to be the strongest predictor. N has consistently and positively been associated with a variety of R/S struggles, including negative appraisals of God (Wilt et al. 2016), anger toward God (Wood et al. 2010), religious crisis (Piedmont 2020), spiritual decline (Wilt et al. 2016), as well as interpersonal, moral, and meaning R/S struggles (Grubbs et al. 2016). Agreeableness (A), a tendency toward compassion and maintaining social harmony (Widiger and Costa 2002), as well as conscientiousness (C), a tendency toward organization and goal-directed activity, are negatively associated with anger toward God (Grubbs et al. 2013; Wood et al. 2010). Extraversion (E), an indicator of interpersonal activity level, interactions, and resulting emotional exuberance (Widiger and Costa 2002), as well as openness (O), a tendency to seek out and appreciate novel experiences, ideas, and values (Widiger and Costa 2002), do not appear to be consistently related to R/S struggles (Grubbs et al. 2016).

The correlational nature of these findings raises a fundamentally important question: are R/S struggles merely a by-product of more basic aspects of emotional distress (i.e., levels of N) or do R/S struggles represent a construct independent of N? Piedmont's (2015) ontological model would argue for the latter, seeing the observed correlational overlap as reflecting the common interest of these measures in emotional distress. Disentangling this overlap thus carries important conceptual implications for both constructs. Research has already shown that the personality dimension of N underlies the experience of most of the categories of distress that are represented in the current diagnostic manuals. Syndromes related to trauma and stress, phobias, impulse disorders, and the personality disorders, among others, have been shown to be linked to high levels of this domain (Miller et al. 2001; Zonderman et al. 1993). However, untangling the partial associations between personality, R/S struggles and psychological distress needs attention and is the focus of this report.

*1.5. The Value of Structural Equation Modeling (SEM) for Clinically Relevant R/S Research*

SEM allows researchers to test high-level hypotheses; specifically theoretically-derived models that define a set of latent dimensions and the hypothesized causal pathways among them. In contrast to individual experiments that focus on a small set of predictors and outcomes, SEM allows for a more comprehensive specification of the constituent elements of a complex phenomenon (Kline 2016). Since the only way to determine causality is with a true experiment, SEM's reliance on mostly cross-sectional, correlational data means that SEM can never prove the existence of causality in actual data. Rather, SEM tests the plausibility of the causal assumptions in the model itself.

SEM does this by deriving a set of expectations of how observed variables ought to relate to one another given the putative causal relations in a model. These expectations are then compared to actual data and congruence determined. For example, if a model says that outcomes Y1 and Y2 are consequents of predictor X1, then we would expect that the observed correlation between Y1 and Y2 be equal to the cross-product of their two path coefficients (i.e., lambdas) from X1. If the observed correlation is the same as the expected one, then support is found for understanding X as having a causal impact on Y1 and Y2. This finding does not mean that a real causal relationship does exist, only that the observed data follow the expectations of this causal model. An experiment would be required to provide definitive proof. The value of SEM is that it allows researchers to specify complete, explanatory models and to determine their viability in real data. SEM also allows researchers to compare the accuracy of several competing models. The model which fits the data best is understood as the probably true model (see Kline 2016 for a full treatment of the interpretive strengths and limitations of SEM).

For the current study, our intent was to examine the potential causal role of religious crisis (RC), a metric of R/S struggle, on emotional well-being. Our hypothesis stated that RC would represent an independent pathway that would potentially impact psychological functioning independent of all the FFM personality domains (Piedmont and Wilkins 2020). Given the ethical sensitivities surrounding any

experimental manipulation of RC, SEM provides a useful alternative approach for addressing causality. While recognizing that any number of "models" can fit a given data set equally well, it is important that the models selected for analysis be determined a priori and reflect meaningful conceptualizations of the phenomenon of interest. In this study, we utilized Piedmont and Wilkins' (2020) ontological model of the numinous and we compared three related models that systematically varied the potential causal nature of RC. As pictured in Figure 1, the first model envisioned RC as an independent predictor of emotional well-being (EWB) from personality. The second model postulated that RC would be merely an outcome from personality; an aspect of emotional distress similar in nature to our EWB variable. The final model viewed RC as being the simple consequence of general mental distress.

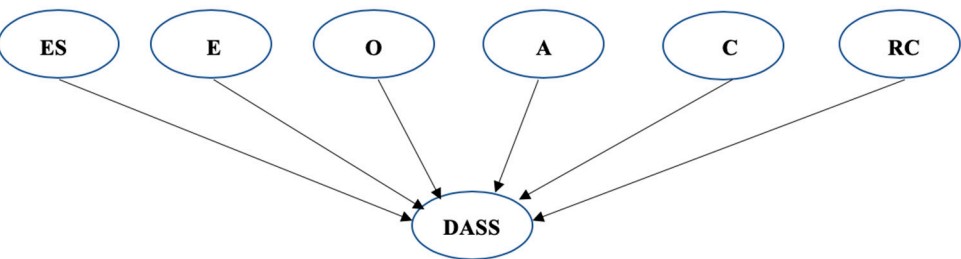

Model 1: RC as Unique Predictor of DASS

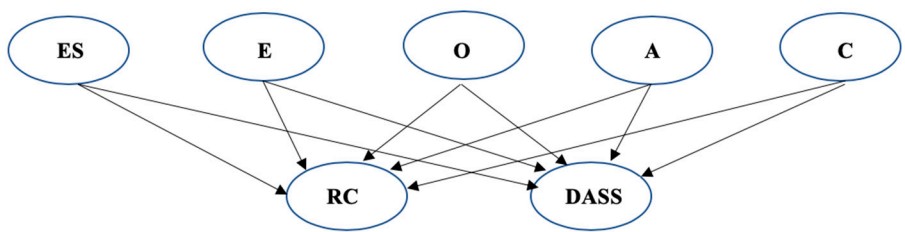

Model 2: RC and DASS as Outputs of Personality

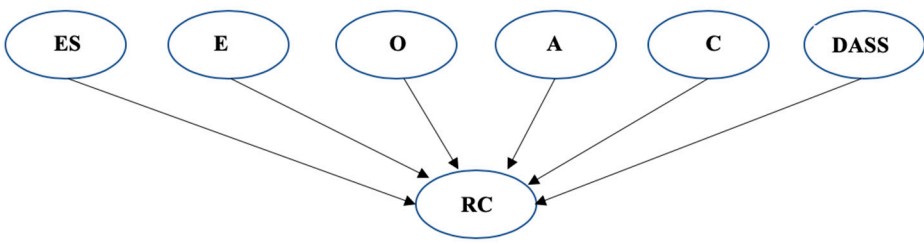

Model 3: RC as an Output of Personality and DASS

**Figure 1.** Proposed structural models linking religious crisis, personality, and psychological distress. Note: ES = emotional stability; E = extraversion; O = openness; A = agreeableness; C = conscientiousness; RC = religious crisis; and DASS = Depression, Anxiety, and Stress Scale-21.

SEM provides a direct comparison of these models. An examination of the various fit statistics for each model will determine which, if any, of the models fits the data best. Being mindful that causality has not been established by these analyses, the best fitting model does acquire conceptual precedence over its rivals in being considered the most accurate representation of the phenomenon being studied.

*1.6. The Current Study*

The purpose of the current study was to directly examine the putative causal role of a measure of W in the resulting experience of psychological distress using SEM. The unique, causal role of W was examined, controlling for the dimensions of the FFM, particularly N. Two issues were addressed: first, it was determined if W does have an independent causal impact on distress, and second, a comparison of the unique contributions of both W and N were made to determine their relative unique contributions.

We predicted that in a real population sample there would be systematic relationships between the measures of the FFM of personality, psychological distress, and R/S struggle. In line with the assumptions of Piedmont and Wilkins (2020), we hypothesized that though the FFM dimensions would provide a robust prediction in our models, RC would provide a substantive, incremental prediction of levels of psychological distress. Likewise, we hypothesized that religious crisis would be an independent causal pathway of explaining psychological distress, and would not be better explained either as an outcome of the FFM or psychological distress itself.

## 2. Method

*2.1. Participants and Procedures*

Before collecting any data from human subjects, the research was approved by Loyola University Maryland's Institution Review Board. The data included in this study were part of a large, multi-study research project on R/S coping in a sample of Amazon's Mechanical Turk (MTurk) workers. Some results of the research program were published in Fox et al. (2017) and Fox and Picciotto (2019). MTurk has become a way to collect data from diverse samples and has been effectively used to investigate R/S constructs as well as psychological distress (Burnham et al. 2018; Engle et al. 2019). Participants read a description of the study, as well as an informed consent, that included a $0.25 incentive for completing a battery of assessments.

In the current study, participants included 226 (72.0%) females and 88 (28.0%) males and the average age of the sample was 38.0 (SD = 13.8) years old. Participants in this study were all affiliated with a religious tradition (*n* = 236, 75.2%) or identified as spiritual but not religious (*n* = 62, 19.7%) or nothing in particular (*n* = 16, 5.1%). Of those who were affiliated with a religion, 217 (69.1%) were Christian, seven (2.2%) were of another faith tradition, five (1.6%) were Hindu, three (1.0%) were Jewish, three (1.0%) were Buddhist, and one (0.3%) was Muslim.

*2.2. Measures*

2.2.1. The Assessment of Spirituality and Religious Sentiments (ASPIRES)

Developed by Piedmont (1999, 2020) this scale is a 32-item measure of numinous motivations. The scale is comprised of two sections. The first section is a measure of religious sentiments, further comprised of two sub facets: religious involvement (RI) and religious crisis (RC). The first nine items measure RI, which captures the extent to which religion is an important part of one's life, including how often one engages in religious activities such as reading sacred scriptures or attending religious gatherings. Responses are recorded using a variety of Likert-type sets. RC captures the degree to which one feels a sense of strain or conflict with the transcendent which Piedmont and Wilkins (2020) referred to as lacking W. For the purposes of this study, we used the RC facet to operationalize the R/S struggle. Piedmont (2020) reported this scale's alpha to be 0.78 normatively with an adequate evidence of construct validity (see also Piedmont and Wilkins 2020). Importantly, this scale has also been shown to be reliable and valid with individuals who are not religious (Toscano et al. 2017). Responses are recorded using a five-point, Likert-type response set ranging from strongly disagree (1) to strongly agree (5). Alpha coefficients for the current study are reported in Table 1.

The second section measures spiritual transcendence. Spiritual transcendence is defined as a universal "capacity of individuals to stand outside of their immediate sense of time and

place and to view life from a larger, more objective perspective. The transcendent perspective is one in which a person sees a fundamental unity underlying the diverse strivings of nature" (Piedmont 1999, p. 988). Spiritual transcendence is comprised of three sub-facets: prayer fulfillment, universality, and connectedness. Prayer fulfillment refers to a sense of satisfaction or joy as a result of personally encountering the transcendent. Universality refers to a belief in the unitive nature of life. Finally, connectedness refers to a belief in ones' participation in a larger human reality which extends beyond generations and groups. Reliabilities for these scales are 0.95, 0.86, and 0.60, respectively (Piedmont 2020). The scales have demonstrated reliability and validity across cultures, religions, languages, and faith orientations (see Piedmont and Wilkins 2020). Responses are recorded using a five-point, Likert-type response set ranging from strongly disagree (1) to strongly agree (5). Alpha coefficients for the current study are reported in Table 1.

**Table 1.** Descriptive statistics by gender and overall alpha reliabilities for study variables.

| Scale | Men (*n* = 88) | | Women (*n* = 224) | | *t* | *α* |
|---|---|---|---|---|---|---|
| | *M* | *SD* | *M* | *SD* | | |
| **IPIP** | | | | | | |
| Emotional Stability | 31.94 | 6.5 | 30.93 | 7.3 | 1.14 | 0.78 |
| Extraversion | 29.32 | 7.6 | 27.27 | 8.3 | 2.00 * | 0.87 |
| Openness | 37.27 | 6.7 | 37.50 | 6.5 | −0.27 | 0.82 |
| Agreeableness | 34.01 | 4.4 | 36.68 | 4.0 | 5.17 *** | 0.83 |
| Conscientiousness | 36.08 | 6.0 | 36.38 | 6.8 | −0.37 | 0.83 |
| **ASPIRES** [A] | | | | | | |
| Religious Involvement | 41.23 | 10.8 | 44.86 | 11.2 | −2.60 ** | 0.88 |
| Religious Crisis | 57.72 | 13.1 | 56.07 | 12.8 | 1.02 | 0.78 |
| Prayer Fulfillment | 50.66 | 7.7 | 54.19 | 6.2 | −4.21 *** | 0.92 |
| Universality | 49.70 | 6.9 | 52.98 | 5.4 | −4.21 *** | 0.77 |
| Connectedness | 51.17 | 6.8 | 52.97 | 7.2 | −2.03 * | 0.40 |
| Total STS | 50.54 | 6.8 | 53.99 | 5.3 | −4.81 *** | 0.89 |
| **DASS** | | | | | | |
| Depression | 5.92 | 5.3 | 5.52 | 5.8 | 0.65 | 0.93 |
| Anxiety | 5.10 | 4.9 | 4.91 | 5.0 | 0.35 | 0.87 |
| Stress | 6.66 | 5.2 | 7.11 | 5.5 | −0.56 | 0.90 |
| **Total Score** | 17.68 | 14.4 | 17.54 | 15.0 | 0.16 | 0.96 |

[A] Scores are presented as T-scores having a mean of 50 and a *SD* of 10 based on normative data provided by Piedmont (2020). Notes: * $p < 0.05$. ** $p < 0.01$. *** $p < 0.001$, two-tailed. IPIP = International Personality Item Pool; ASPIRES = Assessment of Spirituality and Religious Sentiments; STS = Spiritual Transcendence Scale; DASS = Depression, Anxiety, and Stress Scales.

### 2.2.2. The International Personality Item Pool-50 (IPIP)

Developed by Goldberg (1992), the IPIP is a 50-item inventory of the FFM of personality. The scale measures each dimension of the FFM using 10 items, including (a) emotional stability (ES), (b) extraversion (E), (c) imagination (I), (d) openness (O), (e) agreeableness (A), and (f) consciousness (C). Participants read each statement and respond by indicating how it describes them from *very inaccurate* (1) to *very accurate* (5). The IPIP-50 is in the public domain and has demonstrated comparable psychometric qualities to commercial inventories of the FFM (Goldberg et al. 2006; Mlačić and Goldberg 2007). Alpha coefficients for the current study are reported in Table 1.

### 2.2.3. The Depression, Anxiety, and Stress Scale-21 (DASS-21)

Developed by (Lovibond and Lovibond 1995; Brown et al. 1997), the DASS-21 is a brief measure of symptoms common to psychological distress. The instrument consists of three factors (depression,

anxiety, and stress) comprised of seven items emblematic of each form of psychological distress. Participants read through each symptom and respond from *Did not apply to me at all* (0) to *Applied to me very much or most of the time* (3) over the course of the previous week. In a non-clinical sample of the U.S. general population, Sinclair et al. (2012) reported the following means and standard deviations for each scale of the DASS-21: depression = 5.70 (8.2), anxiety = 3.99 (6.3), stress = 8.12 (7.6), and total score 17.80 (20.2). Alpha coefficients for the current study are reported in Table 1.

## 3. Results

### 3.1. Descriptive Statistics

The data were screened for missing data and there were none. The data were then analyzed for multivariate outliers using procedures outlined by Tabachnick and Fidell (2007) and only one case was so identified and removed from the sample. Examinations for univariate outliers and non-normal distributions indicated that the data were appropriate for statistical analyses.

Mean scores on all the study variables obtained separately by gender as well as overall alpha reliabilities are presented in Table 1. Means on the DASS-21 are close to the normative means reported by Sinclair et al. (2012). The ASPIRES scores are presented as T-scores having a mean of 50 and SD of 10 (Piedmont 2020). Scores between 45 and 55 are considered in the average range. As can be seen, scores on the Spiritual Transcendence Scale are all within normal limits. However, scores on RI are in the low range indicating that this sample tends to not involve themselves in specific religious practices. Scores on RC are in the high range, indicating that these subjects tend to feel isolated and punished by God. Significant gender differences were observed on all scales but RC. In each instance, women scored significantly higher than men. Alpha reliabilities were consistent with those found normatively (the low alpha for connectedness was expected (see Piedmont 2004), although its normative value is 0.60). On the other study variables, only one other gender difference was noted, for agreeableness with women scoring higher. Alpha reliabilities for the scores in this sample were all quite acceptable.

### 3.2. Predictive Analyses

The personality and spirituality scales were correlated with the DASS scores and these results are presented in Table 2. As expected, the FFM personality dimension of emotional stability was the largest predictor of all the DASS scores. C and A were also negatively correlated with these outcomes. This personality pattern of associations indicates that individuals with low levels of ES (i.e., high levels of neuroticism), A, and C are most likely to experience higher levels of emotional distress, anxiety, and feelings of depression.

**Table 2.** Correlations between personality, spirituality, religious sentiments and the DASS Scales.

| Predictors | DASS Scale | | | |
|---|---|---|---|---|
| | **Stress** | **Anxiety** | **Depression** | **Overall Score** |
| **IPIP** | | | | |
| Emotional Stability | −0.66 *** | −0.47 *** | −0.57 *** | −0.61 *** |
| Extraversion | −0.10 | −0.05 | −0.17 ** | −0.12 * |
| Openness | −0.11 | −0.13 * | −0.14 * | −0.14 * |
| Agreeableness | −0.19 *** | −0.23 *** | −0.18 *** | −0.21 *** |
| Conscientiousness | −0.37 *** | −0.42 *** | −0.40 *** | −0.43 *** |
| **ASPIRES Scales** | | | | |
| Prayer Fulfillment | −0.24 *** | −0.21 *** | −0.25 *** | −0.25 *** |
| Universality | −0.23 *** | −0.23 *** | −0.20 *** | −0.23 *** |
| Connectedness | −0.09 | −0.07 | −0.11 | −0.10 |
| Total STS | −0.25 *** | −0.24 *** | −0.26 *** | −0.27 *** |
| Religious Involvement | −0.17 ** | −0.16 ** | −0.19 *** | −0.19 *** |
| Religious Crisis | 0.38 *** | 0.33 *** | 0.45 *** | 0.42 *** |

$n$ = 312. * $p < 0.05$. ** $p < 0.01$. *** $p < 0.001$, two-tailed. IPIP = International Personality Item Pool; ASPIRES = Assessment of Spirituality and Religious Sentiments; STS = Spiritual Transcendence Scale.

Interestingly, all of the spiritual transcendence (prayer fulfillment, universality, connectedness) scales, except connectedness, and religious involvement were also significantly negatively related to these outcome variables. Lower levels of spiritual and religious motivations appear linked to the experience of negative affect. Two important questions emerge. First, to what extent do these associations represent unique effects or are merely artifacts of the ASPIRES scales' overlap with the content of the FFM personality scales? While conceptualized to be independent dimensions, the extent to which aspects of A and C are contained in the ASPIRES scales may explain these associations. The second question is whether both spiritual and religious motivations are involved in the experience of negative affect. Are feeling isolated from God and not having a transcendent perspective in understanding one's sense of self and purpose all linked to poor affective regulation? Or again, are the natural overlaps between spirituality and religiousness responsible for all these scales being associated with the DASS? In order to answer this question, a series of four hierarchical multiple regression analyses were conducted, using each of the DASS scales as the outcome variable. In step 1 of the analysis, the five personality dimensions were entered simultaneously. In step 2, the five ASPIRES scales were then entered using a forward entry technique. A partial F-test was conducted to determine whether any of the ASPIRES scales had significant, incremental predictive validity over the FFM domain scores. Furthermore, an inspection of the beta weights will allow for a determination of the relative unique predictive contributions of these scales. The results of these analyses are presented in Table 3.

**Table 3.** Hierarchical multiple regression analyses examining the incremental validity of religious crisis scale in predicting the DASS scale scores.

| Predictor | FFM $R^2$ | RC $\Delta R^2$ | Predictors (Beta) |
|---|---|---|---|
| Stress | 0.47 *** | 0.03 *** | ES (−0.59), C (−0.12), E(0.09), RC (0.17) |
| Anxiety | 0.32 *** | 0.02 ** | ES (−0.36), C (−0.25), RC (0.14) |
| Depression | 0.38 *** | 0.06 *** | ES (−0.42), C (−0.18), RC (0.27) |
| Total Score | 0.44 *** | 0.04 *** | ES (−0.50), C (−0.20), RC (0.21) |

$n = 312$. * $p < 0.05$. ** $p < 0.01$. *** $p < 0.001$. *Note.* FFM = Five Factor Model; ES = emotional stability; E = extraversion; C = conscientiousness; RC = religious crisis.

As can be seen, the personality domains of ES and C were consistent significant predictors of all DASS scales. These findings are consistent with the correlational results. The obtained $R^2$ values indicate that the personality scales explain a moderately strong amount of the variance in the DASS scales. However, it is also important to note that the ASPIRES scales also contributed a significant amount of additional explanatory variances in each of the DASS scales (from 2% to 6% additional variance, a quite substantial amount, Hunsley and Meyer 2003). Thus, numinous motivations do have a unique role to play in the experience of negative affect. An inspection of the beta weights indicates that the RC scale was the only ASPIRES scale to be uniquely related to the outcomes. In fact, the beta weight for RC was in all but one instance the second largest predictor after ES. Both the ES and RC are complementary predictors of negative affect. The other numinous constructs appear to be independent of negative affect.

Regression analysis is able to examine the unique contributions of the predictors relative to one another, capable of distinguishing between the associations that are substantive (i.e., the constructs have direct associations with one another) as opposed to artifactual (i.e., a variable correlates to the outcome because of its association with another variable that predicts). However, these analyses are unable to determine the potential causal precedence of these obtained associations. While RC is a unique, significant predictor of DASS scores, it is not clear why this association exists. Is it because RC has a causal impact on the experience of negative affect or because RC represents another facet of negative affect that is a consequence of the levels of ES? In order to determine this ultimate question, a series of causal models were examined.

### 3.3. SEM Analyses

Figure 1 presents three different models that postulate varying causal roles for RC, consistent with Piedmont and Wilkins's (2020) ontological model of the numinous. Model 1 positions RC as an independent predictor of distress from the FFM. Here, RC is its own causal motivation that works additively with personality to impact levels of emotional dysphoria. The remaining models present RC as an outcome of personality (Model 2) or an outcome of both personality and the experience of emotional distress (Model 3). Using the SEM software Linear Structural Relations (LISREL version 8.73), these different models were examined in the current data set and the results are presented in Table 4.

**Table 4.** Results of the SEM analyses for three causal models.

| Model | $X^2$ | df | $X^2/N$ | RMSEA | SRMR | IFI | CFI | AIC |
|---|---|---|---|---|---|---|---|---|
| 1. | 22.96 | 10 | 2.30 | 0.055 | 0.02 | 0.99 | 0.99 | 92.96 |
| 2. | 48.98 | 11 | 4.45 | 0.093 | 0.06 | 0.98 | 0.98 | 116.98 |
| 3. | 48.15 | 11 | 4.38 | 0.096 | 0.03 | 0.98 | 0.98 | 116.15 |

RMSEA = root mean square error of approximation; SRMR = standardized root mean residual; IFI = incremental fit index; CFI = comparative fit index; AIC = Akaike information criterion.

As can be seen, Model 1, which presents RC as a unique causal predictor of distress, is the best fitting model across all the fit statistics examined, using the criteria proposed by Hu and Bentler (1999) as well as Kline (2016) (i.e., $\chi2/N < 3$; root mean square error of approximation (RMSEA) and standardized root mean residual (SRMR) < 0.05; incremental fit index (IFI) and comparative fit index (CFI) > 0.95). Of particular interest is the Akaike information criterion (AIC) criterion. This index is used when non-nested models are being compared. The AIC examines the parsimony and level of fit for each model in terms that can be directly compared. The model with the lowest AIC is seen as the best fitting model. In Table 4, the AIC analysis indicated that Model 1 fit best. These results support viewing RC as a motivational construct which can potentially impact the psychological equilibrium of individuals, independently of any personality dispositions. The standardized weights (lambdas) for the six contributing factors in this model were as follows: ES: −0.43; E: 0.03; I: 0.06; A: −0.01; C: −0.21; RC: 0.25. RC was the second strongest predictor, contributing 60% as much as ES. No doubt RC is an important variable to include in any study of emotional distress.

## 4. Discussion

Overall, our findings support previous research that has found that personality, particularly the FFM dimensions of N, A, and O, is a robust predictor of both R/S struggles and psychological distress (Grubbs et al. 2016; Piedmont 2020; Wilt et al. 2016; Wood et al. 2010). Moreover, also consistent with previous research, we found that R/S struggles are associated with higher scores of psychological distress and emotional dysphoria (Ano and Vasconcelles 2005; Exline 2013; Exline and Yali 2000; Witvliet et al. 2004; Piedmont forthcoming; Pomerleau et al. 2019). Importantly, our study also extends previous research in some important ways by finding evidence of potential causal pathways that go beyond basic correlations.

While it has long been known that numinous constructs are significantly linked to mental and physical health outcomes (Koenig et al. 2012), there has been little work directed towards understanding why these associations exist beyond simple statistical correlational methods. The psychology of religion and spirituality has been remiss in its efforts at conceptually organizing its area of expertise, from developing a sustainable, consensual definition of its constructs, to creating an ontological understanding of the origins and role of the numinous in our mental lives, to establishing methodological approaches to the development and validation of numinous constructs (Piedmont and Wilkins 2020). The lack of progress in these areas creates serious limitations in providing interpretive contexts for understanding the observed effects for numinous variables, as well as preempting researchers'

abilities to a priori identify and pursue useful directions for future research and construct development (Piedmont 2014).

In the current study, we empirically tested one aspect of Piedmont and Wilkins' (2020) ontological model that outlines the origins and contributions of numinous dynamics within the psychic system. Essentially, this model understands the numinous as a motivational source intrinsic to the cognitive structures of the human species. Our ability to know and understand the world in rich, complex ways contributes to our species' ability to act mostly independently of instinctual scripts. The numinous motivations are seen as representing those psychological qualities that uniquely define the human experience. As such, these constructs are independent of other developed capacities (e.g., the FFM personality domains) which are also common among other species (Gosling et al. 2003; Vonk et al. 2017). Given the unique status of numinous constructs, Piedmont and Wilkins (2020) argued that they represent the essential, organizing qualities that constitute our sense of self (see also Allport 1950).

Since these motivations are uniquely human and exist at the core of our personhood, they should be factors of immense interest to social scientists. The successful satisfaction of these motivations have the potential to confer on individuals a robust sense of resilience and personal abundance in life. These motivations are ultimately synthetic in nature and help to provide a unified sense of self and direction. Incomplete or unsuccessful satisfaction of these motivations can lead to very powerful feelings of dread, emptiness, and emotional distress. As the results of this study demonstrated, when individuals experience serious disturbances in their relationship with the transcendent, they experience a broad sense of emotional distress that can create profound feelings of emptiness and worthlessness. These feelings have been shown to link with characterological impairment (Piedmont et al. 2007) and also with an increased risk for suicidal acting out (Piedmont and Wilkins 2020).

## 4.1. Clinical Implications of the Numinous for Addressing R/S Struggles

The exciting value of the results presented here is that the emotional distress that underlies feelings of existential crisis can be considered as being independent from feelings of psychic distress that are characteristic of the personality dimension of N. While N is a broad band indicator of psychological impairment (e.g., Miller et al. 2001; Zonderman et al. 1993), our findings support previous research that it can no longer be considered as the only pathway to mental distress (e.g., Piedmont et al. 2007). It is easy to see the connection between affective lability and impaired psychological functioning. However, when scores on measures of religious crisis are partialed from N, they continued to be significantly related to psychological impairment. The unique predictiveness of religious crisis (or W) as a predictor of psychopathology raises the exciting possibility that there may be a second independent pathway to impairment that has yet to be charted. This has tremendous importance to the field. First, such a dimension opens the door to discovering new types of pathology that are a consequence of dysfunctionality in one's numinous motivations. Second, it may provide additional insight into extant disorders that appear resistant to standard treatments that address negative affect (e.g., moral injury; Hodgson and Carey 2017). If the numinous has a causal influence on functioning, disruptions in one's numinous motivations impact the core of our personhood and have ramifications for all other aspects of psychosocial functioning (Cheston et al. 2003). Thus, these data suggest the need for a significant revision of the current models of psychopathology and its treatment. For instance, the comparative difference in the predictive effects between N and religious crisis in the current study could be useful in conceptualizing interventions that target underlying factors of pathology more amendable to change. Clinically targeting in-grained personality traits like N is challenging. However, R/S struggles have centuries of theological and spiritual foundation to them, and models of pastoral counseling have capitalized on this centuries-old wisdom and created unique insight into their origin, course, and response to intervention (Maynard and Snodgrass 2015). Moreover, Piedmont and Wilkins (2020) have argued (as have others, e.g., Frankl 1997; Maslow 1971) that there may be a whole new class of mental disorders that need to be discovered that relate to this dimension. Furthermore, it may be possible that

some current disorders which seem resistant to current treatments may be due to a lack of recognition of the presence of numinous distress (e.g., moral injury, body image dysphoria, suicide).

The findings of the current study support the reasoning that R/S are salient to the psychological wellbeing of individuals who may be seeking care and cannot be avoided, and if certain evidence-based practices exist to uniquely address R/S dynamics, then withholding such resources may be grounds for malpractice (Plante 2014). Recent randomized controlled trials of R/S psychotherapy interventions have found that they are capable of causally influencing psychological as well as spiritual functioning (Captari et al. 2018). However, Captari and colleagues found a key difference between the two approaches: when compared to secular psychotherapy, R/S interventions were equally efficacious on psychological outcomes, but produced superior spiritual outcomes. These cause–effect findings (which were experimentally derived) taken together with the current study (which was causally modeled through SEM), support the theory that secular-only psychotherapy models fail to maximally influence spiritual outcomes when practiced apart from interventions which directly address numinous motivations. Thus, neglecting numinous dynamics like religious crisis (or low W) may at the very worst cause significant harm to an individual's psychosocial functioning or at best prevent them from deriving the most benefit from psychotherapy. As Cheston et al. (2003, p. 104) concluded:

> Disturbances in one's relationship with God may exacerbate or even create psychological symptoms. Thus, interventions that do not focus on an individual's spirituality may not be as successful in alleviating the symptoms as treatment that focuses on spirituality because the client will not be reconciled with the higher power object that is central to the presenting problems.

Though many clinicians have been persuaded, there are still large portions of the psychotherapy profession who remain unconvinced of the importance of the numinous for clinical contexts (e.g., Sloan et al. 2001). Recent research has found that only a minority of practicing psychotherapists even assess for R/S dynamics with their patients (Oxhandler and Parrish 2018). However, the adoption of evidence-based practices (EBPs) by the American Psychological Association (APA) has made it possible to assess for R/S issues, and when it is relevant to the patient's values, psychotherapy can be modified to accommodate the patient's unique preferences (APA Presidential Task Force on Evidence-Based Practice 2006). Current efforts to standardized R/S-based treatment manuals have profited from APA's willingness to embrace EBPs (see Religiously-Integrated Cognitive Behavioral Therapy (RCBT) Manuals and Workbooks n.d.). The difference between these previous efforts and the current study is the level of analysis. Treatment manuals often utilize an existing secular psychotherapy model (e.g., Cognitive Behavioral Therapy) which is then adapted to reflect the cultural expressions of a particular R/S. The current study provides an additional step in outlining the potential value of numinous constructs for better understanding human psychological functioning as an organismic drive that is not limited by cultural expression. Understood correctly, this should not be taken to mean that cultural expression is irrelevant to clinical practice. Instead, the cultural expression of R/S has even deeper psychological roots that also need to be incorporated in psychotherapy, and like cultural factors, given room for expression.

While Piedmont and Wilkins (2020) only presented a conceptual model, the data presented here are completely consistent with its assumptions. Certainly, more work needs to be done to replicate these findings and to extend them to more specific types of disorders and then test interventions derived from this model that are targeted at R/S struggles. However, having a useful ontological model for psychologically understanding the numinous provided us with great interpretive depth for the findings from research for clinical practice using these constructs as well as directions for moving forward. Piedmont and Wilkins (2020) provide a clinical case example where the numinous was included as part of the process. How the numinous influenced the client's issues and how its accommodation in treatment enhanced its outcome are documented.

*4.2. Study Strengths, Limitations, and Implications for Future Research*

This study was based on a strong theoretical foundation that clearly delineated the relevant constructs to test how R/S struggles represented unique pathways of psychological maladjustment. Moreover, the study's analytic procedures (SEM analyses) allowed us to empirically compare competing causal models which could disconfirm our presuppositions, adding greater weight to the evidence in support of our hypotheses. However, the study is limited in several ways. First, we only collected one sample—whereas our theoretical model would need replication in different samples representative of other populations (clinical, multicultural, college, youth, etc.) to generalize the findings more broadly. Moreover, we did not use attention checks in this study, which are suggested for MTurk samples. This limitation is mitigated somewhat by the fact that all the scale alpha reliabilities were within acceptable ranges or consistent with the normative data published on the scales. Furthermore, two of our measures (e.g., IPIP and ASPIRES) have balanced keying to control for any acquiescence effects that may arise from such inattention.

This study was only concerned with the antecedent and consequent relationships between R/S struggle and psychological distress (operationalized as a composite of depression, anxiety, and stress), limiting the generalizability of these findings to other forms of maladjustment. Our aim was to provide a straightforward examination of the potential value of the numinous in predicting clinical-type constructs. The positive findings observed here encourage future research to address the potential interactive effects between gender and the numinous, as well as the influence of the treatment type. Future steps ought to include other forms of psychological impairment predicted by R/S struggles (e.g., personality disorders, developmental disorders). Efforts ought now to begin to identify the mechanism(s) of action that connects the numinous to impairment.

Another limitation was that the participants in this study were neither in treatment nor seeking treatment. Scores on the study scales indicate values well within normal limits. While the purpose of this study was to demonstrate the significance of the numinous for psychological distress, future research will need to replicate these findings within clinical samples. With higher scores on both N and RC, the use of clinical samples would more directly test the potential value of the numinous on functioning in a manner most relevant for practitioners. The findings of the current study support such a next step.

Finally, the unique predictiveness of RC over the FFM domains was in the 2–6% range. While this may appear low, it must be kept in mind that the RC scale was being entered on the sixth step of the regression, where these observed variances represent more predictive heft than their nominal values suggest (see Hunsley and Meyer 2003). Nonetheless, it would be unwise to conclude that given the strong predictive role of N in distress, it would be more cost effective if treatment only focused on this dimension. While RC may have a smaller role to play in the development of distress, its consideration may provide key insights and pivotal strategies for overcoming persistent psychological issues that seem refractive to current approaches (e.g., moral injury, suicide, body image dysphoria; Cheston et al. 2003). A focus on the numinous may be easier to manage and intervene clinically than working with the personality domain of N.

**5. Conclusions**

In this study, we have provided and tested a conceptual model for informing our understanding of how numinous constructs may create unique pathways toward psychological distress. Our findings provide a significant step forward for advancing the empirical study and the clinical implications of R/S struggles for psychological functioning. The numinous appears to represent a new set of constructs that can expand our understanding of both resilience and impairment. The potential exists for the discovery of a whole new class of diagnostic conditions and treatment interventions that can significantly move the field forward. We encourage future research to address the limitations we have pinpointed above and to aggressively examine the clinical nosological implications of numinous

constructs. Embedding the numinous within larger conceptual and empirical models provides great promise for moving the social sciences forward in new and productive ways.

**Author Contributions:** Conceptualization, J.F. and R.L.P.; methodology, J.F. and R.L.P.; formal analysis, R.L.P.; investigation, J.F.; resources, J.F.; data curation, J.F. and R.L.P.; writing—original draft preparation, J.F. and R.L.P.; writing—review and editing, J.F. and R.L.P.; visualization, R.L.P.; project administration, J.F. All authors have read and agreed to the published version of the manuscript.

**Funding:** This research received no external funding.

**Acknowledgments:** The authors thank the Dean of the College of Arts and Sciences at Loyola University Maryland for supporting this research through junior faculty research leave. The authors also wish to acknowledge Martin Sherman for his careful review and comments that helped strengthen this manuscript.

**Conflicts of Interest:** Ralph L. Piedmont receives royalties from the sale of the ASPIRES.

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
