# Peer review of "Religious Crisis as an Independent Causal Predictor of Psychological Distress: Understanding the Unique Role of the Numinous for Intrapsychic Functioning"

_religions, doi:10.3390/rel11070329_

Round 1

Reviewer 1 Report

Sorry that this review is so short.

Strengths:

--religious crisis/struggle is a good topic; numinous motivations also interesting; good theoretical framework

--lit review seems to be at right level of depth

`--potentially important connections to personality

--good use of SEM

--potentially important clinical implications

Suggestions:

Please put an additional note on Figure 1 to spell out the variables being abbreviated.

After Table 4, awkward sentence:  “As can be seen in Model 1, which presents…is the best..”

Author Response

We have spelled out any abbreviations in the Tables or Figures and have adjusted the awkward phrasing.

Reviewer 2 Report

I read the paper with great interest until I saw that the compensation rate was a quarter for completing 103 items (by my count). If a participant can complete 8 items per minute, this survey would take about 13 minutes to complete on average, so the compensation is about 2 cents per minute or $1.20 per hour. I didn't see any other non-monetary compensation (e.g., feedback about personality or mental health) listed. This type of exploitation to me is grounds for outright rejection without further consideration. Secondarily to ethical concerns, I have no confidence that the data are reliable given the lack of incentive to provide accurate responses.

Author Response

The study was IRB approved from our University. Part of the reason why we believe this study passed ethical review was because all MTurk studies are open to all workers. When workers survey possible studies to participate in, they are able to see before accessing study informed consents what incentive to participate is available along with an estimate of their time commitment. If either the incentive or the time involved is unappealing, the worker can select another study to participate in without penalization to their status as a MTurk worker. When the worker accesses the study link, they again read and agree to both the incentive and the time involved. We have confidence in the quality of the data because the scales we used in our study include balanced keying items that require participants to provide internally consistent responses. Our alpha values were all within acceptable limits that were consistent with previous research using the scale as well as any normative data available from crowdsourcing, community, and clinical samples.

Reviewer 3 Report

I found this topic quite engaging and information. You tackled a very important  research in the religious and psychology field.

Pay attention to fragmented sentence (see. lines 226-227) and run on sentence (see lines 481-484) to help the grammatical structure of the paper.

Author Response

It appears that the fragmented sentence in lines 226-227 was the result of a missing comma. We have made the correction.

We have fixed the run-on sentence in lines  481-484 and hope that the change helps clarify the meaning of the sentence.

Reviewer 4 Report

This is an inspiring piece of original research which I hope to see published soon. The authors seem very enthusiastic, in sum, I would suggest more soberness, at least in the introduction part. This is my main critical observation. I give some comments and indications first, later on some minor corrections of typos.

Introduction: As a whole, it could be somewhat shorter and more straightforward:

Clear aim of the paper: 38-40: testing the viability of a conceptual model that understands R/S struggles as having a unique, causal impact on psychological distress that is independent of personality. (rather than vice versa)

45-50: this is a very bold affirmation based on one publication of 2000 only (Hill et al. 2000) whereas there is plenty of research of the last 20 years with many (admittedly frequently correlational) studies which maybe have not yet been received by the authors yet. You might better add some more review work in this regard. (cf. below on 1.3)

70-72: from “aspects” (70) and “factors” (71) you change to “qualities” (72), using the terms interchangeably. This seems inadequate to me.

Using the ontological model of Piedmont, you change to the “we”-mode which rhetorically claims to speak for every human being. This may be appropriate sometimes. Reading lines 97-98, however: “In response, we need to find ways to bring meaning, coherence, and depth to this transient life” this sounds more like an appeal; it is not an empirical statement, for many individuals do not look for these ways and seem satisfied enough without such a response or, more interpretively, with suppressing such questions, if “we” like this or not.

98-104: Therefore, notwithstanding my sympathies for ontological convictions, please make clear that these far-reaching claims are either Piedmont’s or yours.

Starting from 104ss is more important as to what empirical evidences you want to work on. Note, however: although FFM is a well-established model of personality, differential psychology would not claim it covers all the psychological functioning or differential aspects of human personality and psychic realities. Or e.g. attachment theory …

Better finally 118-126.

In sum: Try to be more sober in introducing and using (rather than almost zealously “advocating”) Piedemont’s model as a heuristic theoretical basis (there is more zeal then in discussion and conclusions, anyway).

1.3 – R/S struggles and psychological distress – important paragraph; it would be very much worth adding/ mentioning the research (findings) on the type of struggle called spiritual dryness (e.g. by Büssing).

Line 146f you seem to follow the change from correlation to causality by the author you quote. The causality may be the other way, too, however, and should not be neglected. For your goals, you could simply tell here that you intend to go beyond correlational findings (cf. l. 150). 

Line 156: Why “ultimately” – what are the numbers, actually? Or could you do without “ultimately” without losing the content of what you want to report?

Line 158: could you report the strength of this significant mediation? Significance alone is not yet saying too much (cf. negligible significant correlations).

Lines 196-203 rather fit to the discussion and eventually conclusion.

1.5 useful paragraph explaining the rationale of SEM for the purpose of this paper

Figure 1: please explain in the caption what the letters in the figure stand for

1.6 it seems you use RC and W interchangeably. Is this so? If yes, a short explanation is needed – or use one of them only. You had mentioned this before, en passant. For sake of clarity, make it more explicit, please. It ensues only in 2.2.1 more explicitly.

Line 271: the incentive was 25 cents for each participant (only)??

Table 1 – define in the caption what STS stands for, please.

Line 357-359: Do you not consider RC an Aspire Scale? It is positively correlated to DASS. Please correct or explain.

Line 405: explain LISREL 8.73

Line 415-416 – the remark in brackets seems granted (given table 4).

4.2 Good paragraph on limitations and implications for further research.

Line 535: all scale alpha reliabilities – but one with .40 only: Connectedness (Table 1)

Line 551 and others: It is important to make always sure that by “the numinous” you do not intend a transcendent entity or deity, as do religious studies and theologies, but always a psychological construct. Although to me it is clear, it may be useful to emphasize this in advance, e.g. in the introduction.

Typos/ corrections:

Line 27: of the certain – cancel “the”

Line 30: over – do you mean “more than”? It seems so to me.

Line 32: religification – put into quotation marks (“”), it is not an established term yet

Lines 47-48: how is this connected within the sentence: “and the measurement models needed to develop and test them”???

Line 146: cancel “above”

Line 160: more methodological rigorous studies – you mean: methodologically more rigorous studies, don’t you? Or: these studies with more rigorous methods/ with more methodological rigor

Line 185 end: do rather than “does”

Line 422: Enumerate “Discussion” as 4. And following paragraphs correspondingly (4.1 etc).

And 5. Conclusions (not 4)

Author Response

We have uploaded our response in a word document. Thank you for your detailed review!

Round 2

Reviewer 2 Report

I thank the author(s) for the reply regarding research ethics and the validity of data. I encourage the author(s) to read about how the effects of such low rates of compensation undermine psychological research.

Author Response

We will keep this in mind for future research. Thank you.